# Stage Transitions in *Lucilia sericata* and *Phormia regina* (Diptera: Calliphoridae) and Implications for Forensic Science

**DOI:** 10.3390/insects14040315

**Published:** 2023-03-25

**Authors:** Amanda L. Roe, Leon G. Higley

**Affiliations:** 1Biology Program, College of Saint Mary, Omaha, NE 68106, USA; 2School of Natural Resources, University of Nebraska-Lincoln, Lincoln, NE 68583, USA

**Keywords:** insect development, postmortem interval, blow fly, maggot development

## Abstract

**Simple Summary:**

Maggot growth is important in estimating the postmortem interval in cases involving decomposing bodies. In turn, the time in which maggots transition from one stage of development to the next is crucial in determining growth rates. In this study, we examined in detail these transition times for two important blow fly species. Species transitioned between stages following a bell-curve pattern which was not previously known. This new information will be valuable for improving maggot growth determinations and postmortem estimates.

**Abstract:**

Blow fly development rates have become a key factor in estimating the postmortem interval where blow flies are among the first decomposers to occur on a body. Because the use of blow fly development requires short time durations and high accuracy, stage transition distributions are essential for proper development modeling. However, detailed examinations of stage transitions are not available for any blow fly species. Consequently, we examined this issue in two blow fly species: *Lucilia sericata* and *Phormia regina*. Transitions for all life stages across all measured temperatures were normally distributed. Use of probit analysis allowed determination of 50% transition points and associated measures of variation (i.e., standard errors). The greatest variation was noted for the L2-L3, L3-L3m, and L3m-P stage transitions. These results invalidate the notion that largest maggots should be preferentially collected for determining current maggot population stage, and further call into question the relationship between intrinsic variation and potential geographic variation in development rates.

## 1. Introduction

The blow flies (family Calliphoridae) have emerged as the most important single group of insects in forensic entomology [1,2,3]. Their importance arises from their biology, specifically the strong attraction to decay in many species [2,4]. Given their response to decompositional odors coupled with strong flight, it is not surprising that blow flies typically are the first insect colonizers. Indeed, blow flies can arrive at a dead body and lay eggs within literally minutes of death. Consequently, this time of oviposition can be an ideal indicator for estimating time of death. Calculating an estimated time of oviposition itself depends on examining the degree of maggot development at the time of a body’s discovery, and on using temperature and development data to determine when egg laying occurred.

Blow fly development is curvilinear, with a linear section in the center and curves at the low and high temperatures [1]. Typically, most research on blow fly development has focused on the linear portion of the development curve, and few of these studies explicitly report on stage transitions, possibly since the assumption is that larval molting events are relatively synchronous and occur over a short time [5]. However, in a study of *Chrysomya megacephala*, Wells and Kurahashi [6] reported that only the molts from egg to first and first to second stages were “highly synchronized”, each transition occurring in 6 h or less. Kamal [7] reported large ranges (from hours to days) around his presented modal development times, which suggests large transition times where multiple life stages are found together. Most other studies on blowfly development research either treat transitions as a single point or do not discuss transitions at all.

Because there is very little data on stage transitions of blow flies, the associated transition distribution is unknown, which is integral for determining development time within stages. Here, by “transition distribution” we refer to the temporal frequency (counts or proportions through time) at which a new life stage occurs during the molting from one stage to the next.

Mathematically, the issue of transition distribution is identical to that in which any numerical distribution must be determined. Most commonly, frequency distributions are associated with sampling. With blow fly development, the transition distribution not only represents the temporal pattern at which one stage molts into another, but also is a necessary tool for determining stage duration and estimating variation. Broadly, two types of transition distributions are likely. First is the negative binomial (or similar mathematical) distribution that is asymmetric with a peak to the left. With this distribution, stage duration should be calculated by measuring the time from mode to mode between stages. Second is the normal, or Gaussian distribution. If the frequency distribution for stage transitions is normally distributed, then determination of stage duration requires use of the mean (=the peak of the distribution), and stage duration is calculated as the time from peak to peak between stages.

Debates regarding use of mean or mode routinely occur in the forensic entomology literature [1,8], even though no experimental determination of the actual transition distributions exists for all life stages of any blow fly species. Moreover, transition distributions are necessary for measuring variation and attaching confidence intervals around transitions. Finally, transition distributions are important for determining developmental curves, degrees of uncertainty, and sampling protocols, all of which ultimately influence the accuracy of PMI estimations.

The need to determine transition distribution exists, theoretically, in determining the stage duration for any organism that molts. In practice, if the duration of stage transitions are short relative to the total duration of a stage, using single times to represent a transition is adequate. Similarly, if the time of concern is that of the total larval period, or time from egg to adult (as is often the case with agricultural pests), then details on transition distributions are not needed. In contrast, because the use of blow fly development requires short time durations and high accuracy, transition distributions are essential for proper development modeling. We examined this issue in two blow fly species: *Lucilia sericata* and *Phormia regina*.

*Lucilia sericata* (Diptera: Calliphoridae) is a ubiquitous fly that belongs to a group of necrophagous insects that are dependent on decomposing flesh to complete their life cycle [4] and has played major roles in sheep strike and other forms of myiasis, maggot therapy, and death investigations. In the Midwest, *L. sericata* often is the first blow fly to oviposit on a dead body, and for this reason, it is potentially one of the most important species in determining the PMI [2]. The age of maggots when a body is discovered provides a starting point from which to count backward to the time of oviposition, providing an estimate of the duration of the exposure of the body [2]. Developmental research on *L. sericata* has focused on linear portions of the development curve (e.g., [7,9,10]) with temperatures between 16 and 35 °C and no explicit consideration of stage transitions.

In the past, much of the research concerning *Phormia regina* (diptera: Calliphoridae) was dedicated to its role in livestock myiasis [11]. However, with its near complete range across the U.S. (except southern Florida) ([4,12]), *P. regina* has steadily increased its role as a colonizer of human and other animal remains. As such, their role in postmortem interval (PMI) estimations has also increased. Like the situation with *L. sericata,* there are insufficient development data for *P. regina* to allow calculation of stage transition distributions. Among existing studies on *P. regina* development (e.g., [7,10,11,13]), issues include insufficient number of replications, inconsistent temperature regimes, variable sampling protocols, and non-specific life stage intervals.

Ideally, blow fly species would have complete developmental data sets [1]. Unfortunately, developmental experiments are expensive, both in labor and materials, and it would be difficult (if not impossible) to capture and maintain colonies of all blow fly species. Designing developmental studies, therefore, requires choosing species that may or may not share certain life history traits but are of similar importance ecologically or legally. For this study, *P. regina* was chosen because of the species’ increased geographic area, their increasing role in death investigations, and their placement in a different subfamily (Calliphoridae: *Chrysomyinae*) from *L. sericata* (Calliphoridae: *Luciliinae*).

By examining different subfamilies and comparing the data between *P. regina* and *L. sericata,* the overall goal of having concise development data for the majority of the Calliphorids becomes easier if clear patterns emerge. For example, if stage transition times are consistently normally distributed and variation within life stages is similar between species, then it is not unreasonable that overall development patterns may be similar. Thus, our goals in this study were to establish stage transition distributions with confidence intervals across all life stages (egg to adult) and multiple temperatures for *L. sericata* and *Phomia regina.*

## 2. Materials and Methods

### 2.1. Flies

*Lucilia sericata* were obtained from colonies maintained at the University of Nebraska–Lincoln (Lincoln, Nebraska). The colony was established in October 2010, from field-collected insects provided by Dr. Jeff Wells from Morgantown, West Virginia. At the time of research, the colonies achieved 100 generations without addition of new flies.

*Phormia regina* were obtained from colonies maintained at the University of Nebraska–Lincoln (Lincoln, Nebraska). The colony was established in 2011 from field-collected insects in Lancaster County, Nebraska. At the time of research, the colonies achieved 100 generations without addition of new flies. For specific rearing protocols, see Roe and Higley [14].

Adult flies were maintained in screen cages (46 cm × 46 cm × 46 cm) (Bioquip Products, Claremont, CA, USA) in a rearing room at 27.5 °C (±3 °C), with a 16:8 (L:D) photoperiod. Multiple generations were maintained in a single cage, and ca. 1000 adult flies were introduced every 1–2 weeks. Adults had access to granulated sugar and water ad libitum, and raw beef liver for protein and as an ovipositional substrate. After egg laying, eggs and liver were placed in an 89 mL plastic cup which was surrounded by pine shavings in a 1.7 L plastic box. The pine shavings served as a pupation substrate. The 1.7 L box was placed in a I30-BLL Percival biological incubator (Percival Scientific, Inc., Perry, IA, USA) set at 26 °C (±1.5 °C). After eclosion, adults were released into the screened cages.

### 2.2. Incubators

Incubator information has been previously discussed in Authement et al. [14]; pertinent information was revisited here. Incubators were customized, model SMY04-1 DigiTherm^®^ CirKinetics Incubators (TriTech Research, Inc., Los Angeles, CA, USA). The DigiTherm^®^ CirKinetics Incubators have microprocessor -ontrolled temperature regulation, internal lighting, a recirculating air system (to help maintain humidity), and a thermoelectric heat pump (rather than coolant and condenser as is typical with larger incubators and growth chambers). Customizations included the addition of a data port, vertical lighting (so all shelves were illuminated), and an additional internal fan. The manufacturer’s specifications indicate an operational range of 10–60 °C ± 0.1 °C. It is worth noting that a range of ±0.1 °C is an order of magnitude more precise than is possible in conventional growth chambers. Although growth chambers have been shown to display substantial differences between programmed temperatures and actual internal temperatures [15], incubators tested with internal thermocouples in a replicated study showed internal temperatures on all shelves within incubators never varied by more than 0.1 °C from the programmed temperature, in agreement with the manufacturer’s specifications. Given the high level of measured accuracy with programmed temperatures, we were able to use incubators for temperature treatments, which improved our experimental efficiency and helped reduce experimental error.

### 2.3. Experimental Design

The studies with *Lucilia sericata* and *Phormia regina* used the same experimental design. For each species, treatments comprised eleven temperatures (7.5, 10, 12.5, 15, 17.5, 20, 22.5, 25, 27.5, 30, and 32.5 °C) with a light: dark cycle of 16:8. Twenty eggs (collected within 30 min of oviposition) were counted onto a moist black filter paper triangle and placed in direct contact with 10 g of beef liver in a 29.5 mL plastic cup. The cup was placed in a 7 cm × 7 cm × 10 cm plastic container that had 2.5 cm of wood shavings in the bottom for L. sericata, and 2.5 cm of damp sand for *P. regina*. The container was then placed randomly in an incubator. A total of 22 incubators were available for use: the experimental unit was an incubator, temperature treatments were randomized by incubator, and replications were conducted through time to provide sufficient incubators for all treatments. Each life stage (egg–1st stage, 1st–2nd stage, 2nd–3rd stage, 3rd–3rd migratory, 3rd migratory–pupation, pupation–adult) was calculated using Kamal’s [4] data which were converted to accumulated degree hours (ADH) and divided equally into five sampling times (specific times are listed in Table 1 for *Lucillia sericata* and Table 2 for *Phormia regina*). Each sample was replicated 4 times, for a total of 20 samples per life stage (4 replications times 5 sampling periods). In total, there were 120 samples per treatment (4 replications times 5 sampling periods per stage transition times 6 life stage transitions). During each sample time, a container was pulled from each of the four incubators and the stage of each maggot was documented morphologically using the posterior spiracular slits.

During egg hatch, a larva was recorded as 1st stage if they had broken the egg chorion and were actively emerging. Pharate larvae (larvae that have undergone apolysis but not ecdysis) were recorded as the earlier stage (e.g., 3rd stage spiracular slits can be seen beneath the current spiracular slits would be recorded as 2nd stage), since they had not yet molted. Pupariation started when the larva had a shortened body length and no longer projected its mouth hooks when put in the larval fixative KAAD (kerosene-acetic acid–dioxane). There were times when a larva appeared to be entering the puparium stage but would extend its body length and begin crawling if disturbed or placed in KAAD. These larvae were recorded as 3rd migratory. All life stages were preserved in 70% ethyl alcohol. Third and third–migratory stages were fixed in KAAD for 48 h and transferred to 70% ethyl alcohol.

### 2.4. Analysis

As previously discussed, the goal of this experiment was to determine the distributions of stage transitions by temperature for each of 6 transitions. With 11 temperatures and 6 transitions we needed to model 66 relationships for each species. We used 2 regression procedures. First, to determine the appropriate transition distributions, we used TableCurve 2d, version 5.01 (SYSTAT Software Inc., San Jose, CA, USA, http://www.sigmaplot.com/products/tablecurve2d/tablecurve2d.php), accessed on 1 July 2022. and Prism, version 6.02 (GraphPad Software, Inc., La Jolla, CA, USA, http://www.graphpad.com/scientific-software/prism/). Here, we fit one of 4 functions (specifically, a regressed proportion (percentage) in stage versus time, at each temperature tested). Variables were *y* = proportion in stage, *x* = time (degree days), and *a, b, c*, and *d* = regression coefficients. The equations used were the following:A Gaussian equation (a standard normal curve):
$$y=a\exp\left[-\frac{1}{2}{\left(\frac{x-b}{c}\right)}^{2}\right].$$

2.A modified Gaussian equation (a form of Gaussian curve with a plateau at 100%):


$$y=a\exp\left[-\frac{1}{2}{\left(\frac{\left|x-b\right|}{c}\right)}^{d}\right].$$


3.A cumulative Gaussian equation (a form of the Gaussian curve used for adults to model a sigmoidal increase to a plateau):


$$y=\frac{a}{2}\;\left[\;1+erf\left(\frac{x-b}{\sqrt{2}c}\right)\right].$$


4.A reversed cumulative Gaussian equation (a form of the cumulative Gaussian equation used for eggs, to model a sigmoidal decrease from a plateau):


$$y=\frac{a}{2}\;\left[\;1-erf\left(\frac{x-b}{\sqrt{2}c}\right)\right].$$


Cumulative forms of the equations were needed to model the transitions from egg or to adult. For the larval and pupal stages, the distinction between fitting a Gaussian or modified Gaussian equation usually depended on length of time in stage. Because longer lasting stages often had a plateau when all individuals were in the same stage between transitions, the modified Gaussian relationship was more appropriate. Fitting these relationships provided evidence for the mathematical distribution of individuals during stage transitions.

A different regression procedure was needed to determine the duration of individual stages (50% of L1 to L2, for example). Various approaches could be used, for example, determining the time from peak of one stage to peak of the next. However, we used the time between 50% transition into a stage to 50% transition out of a stage. We made this choice because we can determine a standard error in the 50% transition point, which is not always possible with determining peaks. Determination of the 50% transition point itself is straightforward using a probit model, with the probit choices of being in the first stage or the next. Through probit modelling, it is possible to determine any desired % transition and the associated variation. Probit models were constructed with Prism 6.02.

For all regression analyses, the data were examined closely to determine their propriety for inclusion in analysis. In a few instances, individuals were sampled with extraordinarily extended durations. These were treated as outliers and excluded from analysis (indicated in Appendix A).

## 3. Results

### 3.1. Lucilia sericata

Data for *Lucilia sericata* can be found in Appendix A and all calculations are contained in Appendix A. Data from 7.5 °C were not analyzed because there was very little egg development and no larval eclosion. In all 66 stage transitions examined, transition frequencies were normally distributed (Figure 1 shows the normal distribution of the transitions periods). Gaussian curves were used on life stages that had little or no plateauing, whereas modified Gaussian curves were a better fit for plateaued data (i.e., pupation). There were large spreads in transition times, particularly during the later life stages (third, 3m, and pupation) for all temperatures reported (Figure 1), covering a period of hours (egg, first, second stages) to days (third, 3m, pupation).

Probit analysis indicated 50% transition times (Figure 2). There was large variation at 10.0 and 12.5 °C, the majority due to high mortality rates (Figure 2). The effect of mortality can also be seen at 30.0 and 32.5 °C (Figure 2), with few individuals reaching 100% of the early life stages. However, there was still enough resolution to determine the means for those temperatures (Table 3).

### 3.2. Phormia regina

Data for *Phormia regina* can be found Appendix A and calculations are contained in Appendix A. Data from 7.5 °C were not used because there was no egg development. Experiments at 10 °C were stopped at 400 h (16.7 days) due to complete egg and first stage mortality. The remaining stage transitions were all normally distributed (Figure 3 shows the normal distribution of the transitions periods). The later life stages, particularly L3m and pupation, had the largest variability, with large error bars on most of the samples. Starting at 12.5 °C, the pupation stage was broad, covering hundreds of hours, which was also true at 32.5 °C, where the entire life cycle was complete in 250 h.

The method of calculating sampling times was not as accurate in this species as compared to *L. sericata* for the L3m stage (Figure 4). Of all the L3m samples taken, few were at the required time within the stage, with 6 of the 10 temperatures never sampling at 100%. The most pronounced data reduction can be observed at 15.0 °C. Figure 4 illustrates this issue more clearly by the almost vertical slope and <100% in stage observed at the L3m locations. However, there were enough data to determine the means for all temperatures (Table 4).

## 4. Discussion

### 4.1. Lucilia sericata

Just as cooler temperatures yield longer development times [1,4,8,9], cooler temperatures also produce longer transitions times, as well as larger confidence intervals. Since blow flies are poikilothermic, extreme temperatures interfere with biological processes including metabolism, movement, and the regulation of growth hormones. This leads to developmental variation at these temperatures, which was observed at 10.0 and 12.5 °C and 30.0 and 32.5 °C. Additionally, there also appears to be an inherent variability in development, as shown by the wide transition times. The insects used in these experiments had been inbred through 100+ generations, making it improbable the transition times were due to underlying genetic variability. Long transition times could be associated with suboptimal rearing conditions; however, other evidence (e.g., total development time, larval size, and survivorship) is not consistent with this explanation. Consequently, the observed variation in stage transition by individual maggots, which gives rise to long transition times, seems to be an intrinsic trait in *L. sericata.* Possibly, this underlying variability is a means for individuals to survive in an ephemeral environment. Carcasses are rarely in the environment for very long and can be found in a wide expanse of temperatures, humidity, and locations. The ability to complete a life cycle over a broad range of conditions could reduce intraspecific competition, making the survival of the species more likely.

When we examine the data that can be most strongly compared (since we are not using the exact temperatures), our results are similar to those in the modes reported in Kamal [7] (Table 5), with a temperature difference of 26.7 °C (Kamal [7]) versus 27.5 °C. We can compare more directly as percent of larvae within a stage, which eliminates trying to compare mode to mean measurements (Table 6). Most variability is observed in the later life stages (third migratory and pupation), which is where most transition variability is found. There is also a strong comparison between our data and the limited data from Ash and Greenberg [9], with 27.0 °C (A & G) versus 27.5 °C (here). When we compare our calculated means, six of nine are within 2 SD and the remaining three are within 3 SD (Table 6), with the differences shown in Table 7.

Both Kamal [7] and Ash and Greenberg [9] used continuous lighting during their development studies, which could account for some of the variation, since it has been shown light regimes can affect development [16]. Interestingly, Kamal’s population of *L. sericata* was collected from Pullman, WA, Ash and Greenberg’s population were collected from Chicago, IL, and ours were collected from Morgantown, WV. There was also a large time difference in population examinations: 56 and 39 years, respectively. The fact that these data sets produce similar results raises a question about geographic variation and its impact on development. If geographic variation caused a considerable difference in development times, the transitions among the three data sets should have been significantly different, but they were not.

### 4.2. Phormia regina

*Phormia regina* is known for its preference for cooler temperatures [12]. However, this species did not mature past the first stage at 10.0 °C in this study and appeared to have just as much difficulty maturing to adults at 12.5 °C and 15.0 °C as *L. sericata*. Unlike data in Byrd and Allen [11], there was an egg hatch at 10.0 °C, but since there was no development past the L1 stage, *P. regina*’s biological minimum temperature is between 10.0 °C and 12.5 °C.

While there are some blow fly species, specifically *Calliphora vicina*, that can complete their life cycle at temperatures below 10.0 °C [17], *P. regina* does not appear to be one of them. Surprisingly, the transition rates at 30.0 °C and 32.5 °C were not adversely affected by temperature. This could be partially explained by mortality rates. There was not an increase in mortality as the temperatures increased, unlike with *L. sericata*, where mortality did increase at the higher temperatures (less mortality equals more data to analyze).

There is similar variation between the two species during the later life stages transitions. Both *P. regina* and *L. sericata* have large variation during the L3m and pupation stages. Unlike *L. sericata*, however, *P. regina*’s pupal stage does not begin to plateau until 22.5 °C and 6 out 10 curves (Figure 4) calculated for L3m did not reach 100%. This could be an artifact of the sampling protocol, where the method of calculating sampling times was not as accurate in this species. Therefore, although the sampling times were divided into five equal times, those times did not align with the transitions during the later life stages. Regardless, as with *L. sericata*, the observed variation in stage transition by individual maggots seems to be intrinsic in *P. regina*. Because all necrophagous species rely on ephemeral resources, it could be expected that those with more intrinsic variation are most likely to survive over a broad range of environmental factors and the trait is shared between blow fly subfamilies.

Because of the vast differences in methodologies and temperatures studied, it is difficult to directly compare data sets. Our results do compare favorably with those of Kamal [7] when we compare percent of larvae within a stage (Table 8) at similar temperatures (26.7 °C (Kamal [7]) and 27.5 °C). Converting transition times to percentages eliminates the need to attempt to compare mode to mean measurements (Table 9). Most variability is observed in the L3m stage in both data sets, which coincides with where we detected the majority of transition variation. Our data also favorably compare to those of Byrd and Allen [11] at the higher temperatures (25.0 and 30.0 °C), with our transition times for all life stages fitting within their given ranges. Our transition times are considerably greater than their reported ranges at the cooler temperatures (15.0 and 20.0 °C).

Differences between these data and those of Byrd and Allen [11] could be a result of methodology. Byrd and Allen used 400 eggs per subsample (with a total of three samples). It has been proposed that maggots in large masses feed more efficiently due to external digestive enzymes [3]. In this case, faster feeding and subsequent digestion could push development forward and could explain the faster development times reported by Byrd and Allen [11].

As with *L. sericata*, the question of whether geographical variation significantly impacts development rates is raised. Kamal’s [7] population of *P. regina* was obtained from Pullman, WA; Byrd and Allen’s [11] population was obtained from Florida, and ours was collected from Lincoln, NE. Although there are differences between the data sets, it is unlikely that the variation was caused by the geographic differences in the fly populations due to them being similar despite methodological differences between the experiments and the high intrinsic variation seen within our highly inbred population.

While this study was very similar to the one conducted with *L. sericata*, the result is the same: accurate transition data leads to more accurate developmental data, which leads to more accurate development models. Models can accommodate the linear and curvilinear areas of development and can be used over a series of temperatures. As discussed with *L. sericata*, collecting development data for as many forensic species as possible is imperative. Comparisons between multiple, comprehensive data sets allows similarities, differences, and patterns to be recognized, increasing our knowledge of basic development biology and the variables that affect it.

### 4.3. Forensic Implications of Findings

Because transitions were normally distributed, the assumption that stage transitions are rapid with a long fall off is disproved. Consequently, the current practice/recommendation of collecting the largest maggots from a mass (e.g., [18]) is incorrect. By using a small sample of the largest maggots, an underrepresentation of the actual cohort age is initiated, both through sampling error and failure to properly represent transition distribution. While maggot size has often been used as an age determinate, there is a vast difference in maggots reared in a laboratory (where food and/or competition are not limiting factors) versus a carcass (where food and competition are limiting factors). Since nutrition has a direct impact on larval size, it is difficult to control for size among “unknown” age samples, such as those commonly found in death/myiasis investigations. Even with controlled populations, during the earlier life stages, stage is “more effective than size for estimating the [larval] age” [6].

Because the calculation of degree day requirements by stage depends on the determination of when one stage transitions into another, our results highlight a potential source of error in determining degree-day requirements by stage. Additionally, the large range of transition values for the L3-L3m, L3m-P, and P-A stage transitions suggest that degree-day determinations for these stages (i.e., L3, L3m, and P) are particularly given to high variability. One approach for addressing this variation would be to explicitly consider the proportion of individuals in multiple stages and using these to determine where the population is on a stage transition curve. Obviously, for such an approach to be viable, detailed transition curves must be available and samples of maggot must accurately represent the underlying population.

A final implication we have previously mentioned is the controversial question of ways to measure geographic variation between populations of a given species. Given the high degree of variation we see in transition times in this study (with species in different subfamilies), unless the variation in transition time is explicitly considered, it is easy to misconstrue variation associated with transitions with geographic variation.

## Figures and Tables

**Figure 1 insects-14-00315-f001:**
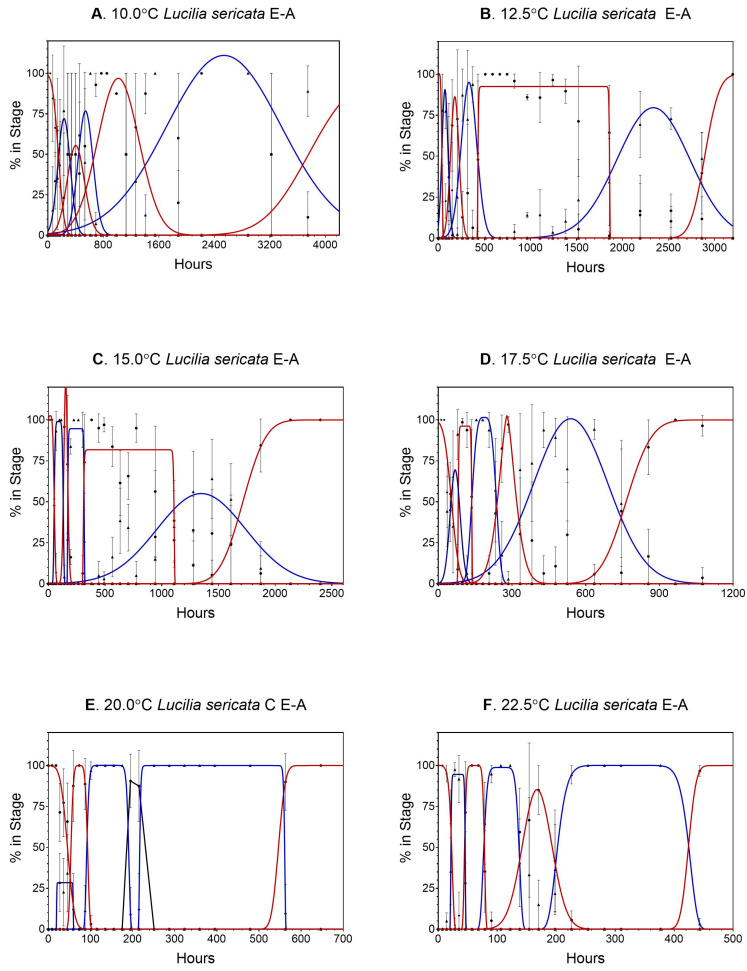
(**A**–**F**) Gaussian and modified Gaussian curves (those curves with plateau) fit to all life stages (egg to adult) of *Lucilia sericata* for temperatures 10.0 to 32.5 °C. (**A**–**J**), respectively. Models were not constrained, so occasionally the best fit curve can exceed 100%. Different line colors and black dot and triangles are used to distinguish adjacent curves. (**G**–**J**). Gaussian and modified Gaussian curves (those curves with plateau) fit to all life stages (egg to adult) of *Lucilia sericata* for temperatures 10.0 to 32.5 °C. (**A**–**J**), respectively. Models were not constrained, so occasionally the best fit curve can exceed 100%. Different line colors and black dot and triangles are used to distinguish adjacent curves.

**Figure 2 insects-14-00315-f002:**
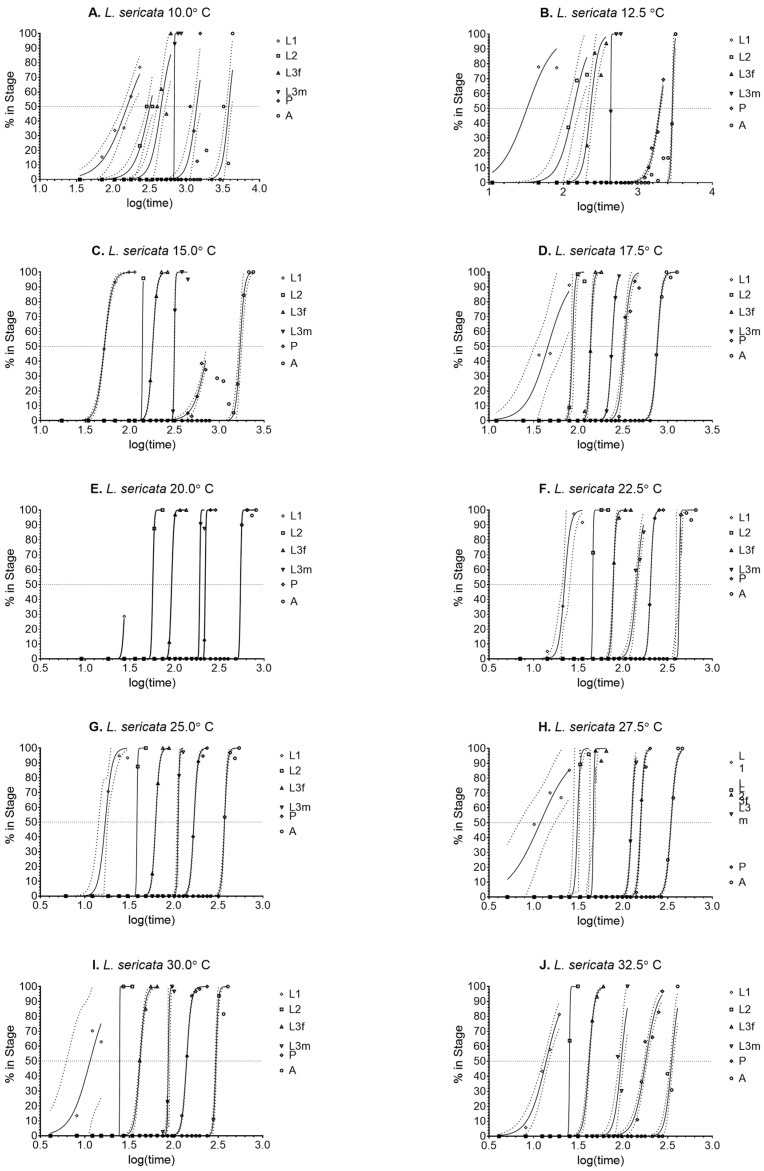
Probit analysis curves used to determine 50% of the transition population for *Lucilia sericata* at 10.0–32.5 °C (Figure 2**A**–**J**, respectively). Confidence intervals are represented by dotted lines.

**Figure 3 insects-14-00315-f003:**
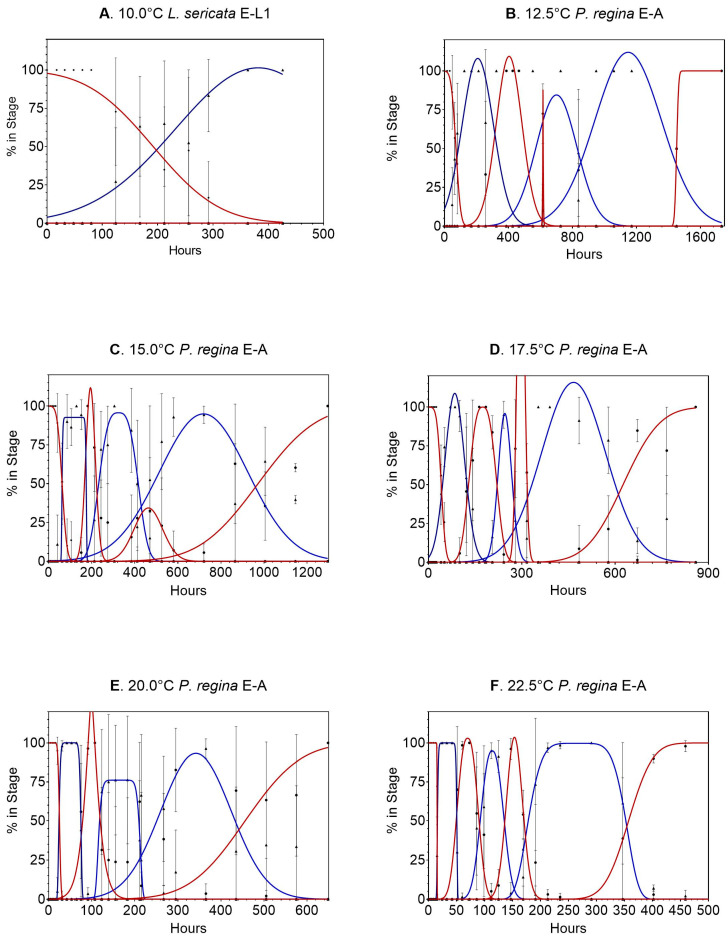
(**A**–**F**) Gaussian and modified Gaussian curves (those curves with plateau) fit to all life stages (egg to adult) of *Phormia regina* for temperatures 10.0 to 32.5 °C (Figure 1A–J, respectively). Models were not constrained, so occasionally the best fit curve can exceed 100%. Different line colors and black dot and triangles are used to distinguish adjacent curves. (**G**–**J**) Gaussian and modified Gaussian curves (those curves with plateau) fit to all life stages (egg to adult) of *Phormia regina* for temperatures 10.0 to 32.5 °C (Figure 1A–J, respectively). Models were not constrained, so occasionally the best fit curve can exceed 100%. Different line colors and black dot and triangles are used to distinguish adjacent curves.

**Figure 4 insects-14-00315-f004:**
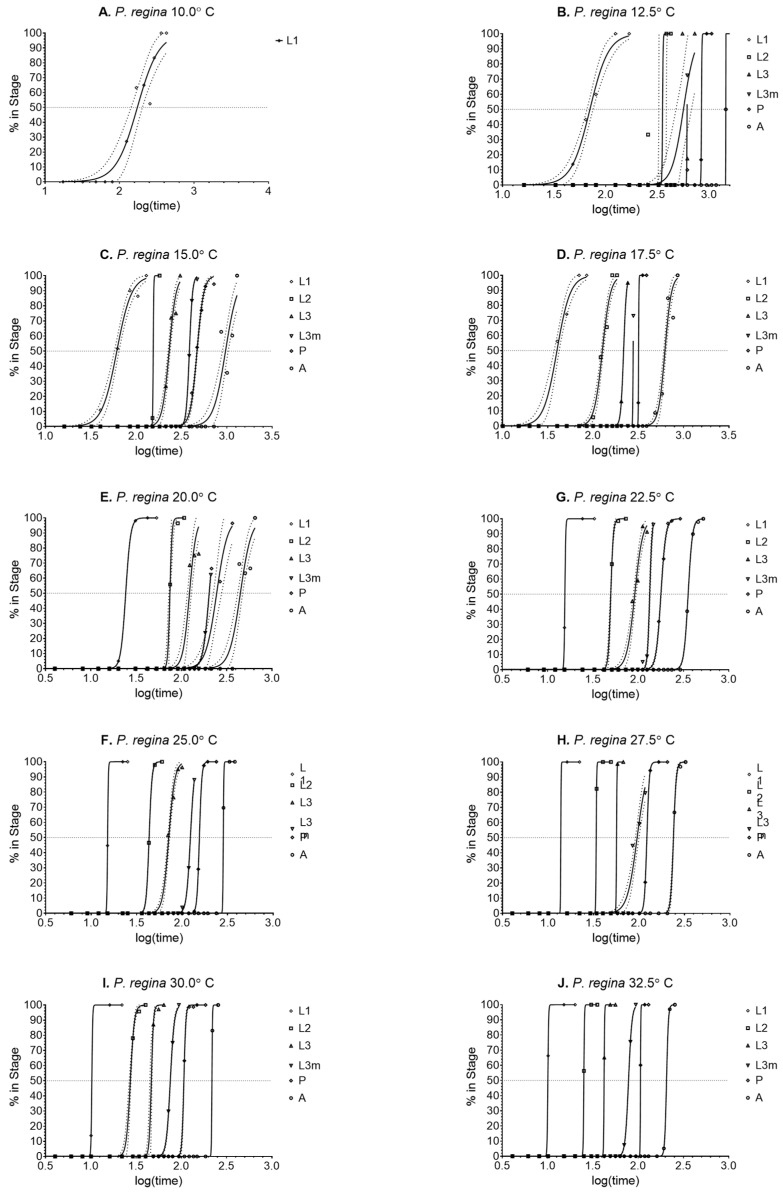
(**A**–**E**). Probit analysis curves used to determine 50% of the transition population for *Phormia regina* at 10.0–32.5 °C (**A**–**J**, respectively). Confidence intervals are represented by dotted lines.

**Table 1 insects-14-00315-t001:** Sample times for *Lucilia sericata* were calculated by converting the minimum and maximum data reported in Kamal [4] into accumulated degree hours (ADH). The ADHs were calculated for each life stage and sampling temperature, converted back into hours and divided into 5 equal sample times.

	Temperature °C										
**Life Stage**	7.5	10.0	12.5	15.0	17.5	20.0	22.5	25.0	27.5	30.0	32.5
Egg–1st	35	35	35	17	12	9	7	6	5	4	4
1st–2nd	56	56	56	28	19	14	11	9	8	7	6
2nd–3f	79	79	79	39	26	20	16	13	11	10	9
3f–3m	143	143	143	71	48	36	29	24	20	18	16
3m–Pupal	335	335	335	167	112	84	67	56	48	42	37
Pupal–Adult	527	527	527	263	176	132	105	88	75	66	59

**Table 2 insects-14-00315-t002:** Sample times for *Phormia regina* were calculated by converting the minimum and maximum data reported in Kamal [4] into accumulated degree hours (ADH). The ADHs were calculated for each life stage and sampling temperature, converted back into hours and divided into 5 equal sample times.

	Temperature (°C)										
**Life Stage**	7.5	10.0	12.5	15.0	17.5	20.0	22.5	25.0	27.5	30.0	32.5
Egg–1st	16	16	16	8	5	4	3	3	2	2	2
1st–2nd	44	44	44	22	15	11	9	7	6	6	5
2nd–3f	63	63	63	31	21	16	13	10	9	8	7
3f–3m	111	111	111	55	37	28	22	18	16	14	12
3m–Pupal	281	281	281	141	94	70	56	47	40	35	31
Pupal–Adult	441	441	441	221	147	110	88	74	63	55	49

**Table 3 insects-14-00315-t003:** Mean time (hours) for *Lucilia sericata* to reach 50% of the population for each life stage and temperature.

	Temperature °C										
**Life Stage**	7.5	10.0	12.5	15.0	17.5	20.0	22.5	25.0	27.5	30.0	32.5
Egg–1st	N/A	186	41	52	47	28	21	17	18	12	14
1st–2nd	N/A	407	136	131	79	59	46	36	31	24	24
2nd–3f	N/A	463	231	179	137	92	78	63	49	42	36
3f–3m	N/A	683	431	314	237	193	141	111	124	85	95
3m–Pupal	N/A	2179	1935	1206	326	217	202	168	158	139	180
Pupal–Adult	N/A	4011	2896	1710	760	554	424	371	344	297	353

**Table 4 insects-14-00315-t004:** Mean time for *Phormia regina* to reach 50% of the population for each life stage and temperature (hours).

	Temperature (°C)										
**Life Stage**	7.5	10.0	12.5	15.0	17.5	20.0	22.5	25.0	27.5	30.0	32.5
Egg–1st	N/A	173	70	61	40	24	16	15	14	10	10
1st–2nd	N/A	N/A	354	154	127	75	50	43	34	27	25
2nd–3f	N/A	N/A	564	231	218	123	92	71	57	47	42
3f–3m	N/A	N/A	612	521	277	202	134	123	95	76	78
3m–Pupal	N/A	N/A	843	469	319	245	178	155	122	107	106
Pupal–Adult	N/A	N/A	1450	966	622	447	356	283	239	218	206

**Table 5 insects-14-00315-t005:** Comparison between Kamal [7] and Roe and Higley of *Lucilia sericata* as percent in stage.

	Temp	Egg	L1	L2	L3f	L3m	P
R and H	27.5	5.2%	3.7%	5.4%	22.0%	9.7%	54.1%
Kamal	26.7	5.2%	5.7%	3.4%	11.5%	25.9%	48.3%
**Difference**		0.0%	−2.1%	1.9%	10.5%	−16.2%	5.8%

**Table 6 insects-14-00315-t006:** Comparison between Ash and Greenberg [9] and Roe and Higley of *Lucilia sericata* mean transition times.

Transition Stages	Mean of Transition Time in Hours					
	A & G 1975	R & H	A & G 1975	R & H	A & G 1975	R & H
	19.0 C	20.0 C	27.0 C	27.5 C	35.0 C	32.5 C
E-L1	29.4	28.1	14.4	17.9	10.2	13.8
L3m-P	691.2	216.8	194.4	157.6	333.6	180.0
P-A	1312.8	553.7	384.0	343.5	295.2	352.5

**Table 7 insects-14-00315-t007:** Difference in transition means observed here for *Lucilia sericata* as proportion of Ash and Greenberg [9] standard deviations.

Transition Stages	Difference in Transition Means (A&G (1975)—Presented Data) as Proportion of A&G Standard Deviations		
	19 and 20 C	27 and 27.5 C	35 and 32.5 C
E-L1	0.6	2.7	1.9
L3m-P	3.9	0.5	0.9
P-A	2.5	0.3	1.0

**Table 8 insects-14-00315-t008:** Comparison between Kamal [4] and Roe and Higley of *Phormia regina* development means and modes.

		E-L1	E-L2	E-L3f	E-L3m	E-P	E-A
R and H	mean	13.8	33.9	56.6	95.3	122.1	239.2
Kamal	mode	16	34	45	81	165	309
**Difference**		−2.2	−0.1	11.6	14.3	−42.9	−69.8

**Table 9 insects-14-00315-t009:** Comparison between Kamal [4] and Roe and Higley of *Phormia regina* as percent in stage.

	Temp	Egg	L1	L2	L3f	L3m	P
R and H	27.5	6.1%	8.9%	10.1%	17.2%	11.9%	52.0%
Kamal	26.7	5.2%	5.8%	3.6%	11.7%	27.2%	46.6%
**Difference**		0.9%	−3.1%	6.5%	5.5%	−15.3%	5.4%

## Data Availability

Data for this study are available in Appendix A.

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
