# Peer review of "Stage Transitions in *Lucilia sericata* and *Phormia regina* (Diptera: Calliphoridae) and Implications for Forensic Science"

_insects, 2023, doi:10.3390/insects14040315_

Round 1

Reviewer 1 Report

The results of this study have potential value for using forensic entomological data to estimate PMI, deserves recommendation. However, some contents of the manuscript need be carefully revised before it can be considered for publication. Following are some comments you should consider while revising your manuscript.

Line 310: Please clearly mark the meaning of each curve in Figure 1. Which is the Gaussian curve? Which one is the modified Gaussian curve? The same goes for Figure 3.

Line 167-176: Table 2, please resize the table appropriately to make the manuscript look more comfortable after publication. The same goes for Table 3 and Table 5.

Line 323-324: Are there other similar studies that support this idea (cooler temperatures yielded longer transitions times)? Please provide relevant references if available.

Author Response

Regarding points highlighted by review #1

  1. text was added to legend indicating the modified Gaussian curves were those that showed a plateau
  2. tables were moved left which should eliminate the problem with line numbers bleeding through
  3. I did a slight rewrite and added examples references regarding cool temperatures and extended development. To the best of our knowledge, cool temperatures and extended transitions have not been previously documented (although they have probably been assumed to follow trends for development times).

Reviewer 2 Report

This is a well-written manuscript based on good research thus needs only a few clarifications and adjustments. 

Author Response

Response to points in pdf:

  1. Ouch -- fixed the misspelling
  2. I think the disagreement here must involve the difference between developmental studies (there are many) and studies focused on transition times explicitly. To the best of our knowledge there are no other studies that have focused explicitly on transitions (which is why some incorrect assumptions regarding transition frequencies have persisted). If we are wrong in this, please indicate a reference or references.
  3. rewrote to indicate in the Midwest
  4. rewritten
  5. I'm not clear on this comment, but I did a minor rewrite to clarify
  6. liver - it would dry on the upper surface but remained moist underneath and larvae continued to feed. For larger maggots the saliva (salivary enzymes I suppose) re-wet the dry area and the group of maggot continued to feed. I suspect high humidity in our rearing containers may also explain the difference we observed versus your experience
  7. I added the following explanation to the experimental design section: "A total of 22 incubators were available for use: the experimental unit was an incubator, temperature treatments were randomized by incubator, and replications were conducted through time to provide sufficient incubators for all treatments. "
  8. we used Kamal's data because in Roe and Higley 2015 we found the closest agreement between our developmental data and that of Kamal's (despite the limitation of Kamal only providing single temperature - development points) This mirrors some of our practical experience in case work in which estimates based on Kamal proved closer than other models.
  9. as explained in the methods, the containers were removed for each sampling time (destructive sampling). I modified the Methods to make this clearer. In total, for each treatment there were 4 replications times 5 sampling periods per life stage transition times 6 stage transitions for 120 total per treatment.
  10. mortality is discussed later in the discussion section
  11. variable definitions added (good catch -- thanks!)
  12. explanation added to figure legends (models weren't constrained to 100%)
  13. fixed